# AN EMPIRICAL STUDY OF NEURAL CONTEXTUAL BANDIT ALGORITHMS

## ABSTRACT

Recent advances in representation learning have made significant influences on solutions of contextual bandit problems. Neural bandit algorithms have been actively developed and reported to gain extraordinary performance improvement against classical bandit algorithms in numerous papers. However, there lacks a comprehensive comparison among the existing neural bandit algorithms, and it is still not clear whether or when they can succeed in complex real-world problems. In this work, we present an inclusive empirical study on three different categories of existing neural bandit algorithms on several real-world datasets. The results show that such algorithms are highly competitive against their classical counterparts in most cases, however the advantage is not consistent. The results also reveal crucial challenges for future research in neural bandit algorithms.

## 1 INTRODUCTION

In recent decades, contextual bandit algorithms have been extensively studied (Langford & Zhang, 2007; Chu et al., 2011) for solving sequntial decision-making problems. In such problems, an agent iteractively interacts with the environment to maximize its accumulated rewards over time based on the given context. The essence of contextual bandits is to balance exploration and exploitation under uncertainty. In practice, contextual bandit algorithms have wide applications in real-world scenarios, including content recommendation (Li et al., 2010; Wu et al., 2016), online advertising (Schwartz et al., 2017; Nuara et al., 2018), and mobile health (Lei et al., 2017; Tewari & Murphy, 2017).

Linear contextual bandits, which assume the expected reward is linearly related to the given context features, have been extensively studied in literature (Auer et al., 2002; Rusmevichientong & Tsitsiklis, 2010; Dani et al., 2008; Abbasi-Yadkori et al., 2011; Chu et al., 2011). Though linear contextual bandit algorithms are theoretically sound and succeed in a number of real-world applications, the linear assumption fails in capturing non-linear relations between the context vector and the reward. This motivates the study of generalized linear bandits (Li et al., 2017; Faury et al., 2020; Filippi et al., 2010) and kernelized bandits (Krause & Ong, 2011; Chowdhury & Gopalan, 2017; Valko et al., 2013). Recently, deep neural networks (DNN) (LeCun et al., 2015) have been introduced to learn the underlying reward mapping directly. (Riquelme et al., 2018) developed NeuralLinear, which applied a Bayesian linear regression on the feature mappings learned by the last layer of a neural network and get the approximation of the reward via Thompson Sampling. (Zahavy & Mannor, 2019) extended NeuralLinear by adding a likelihood matching mechanism to overcome the catastrophic forgetting problem. (Xu et al., 2020) proposed Neural-LinUCB by performing exploration over the last layer of the neural network. NeuralUCB (Zhou et al., 2020), NeuralTS (Zhang et al., 2020) and NPR (Jia et al., 2021) explore the entire neural network parameter space to obtain nearly optimal regret using the neural tangent kernel technique (Jacot et al., 2018).

All the proposed neural contextual bandit algorithms reported encouraging empirical improvement compared to their classical counterparts or a selected subset of neural contextual bandit algorithms. However, there still lacks a horizontal comparison among the neural contextual bandit solutions on more comprehensive real-world datasets. We argue, for practical applications, it is important to understand when and how a neural contextual algorithm better suits a specific task. In this work, we provide an extensive empirical evaluation on a set of most referred neural contextual bandit algorithms on nine real-world datasets: six K-class classification datasets from UCI machine learning datasets (Dua & Graff, 2017), one learning to rank dataset for web search , and two

logged bandit dataset for online recommendations . We choose LinUCB as a reference linear bandit algorithm against six selected neural contextual bandit algorithms: NeuralLinear , NeuralLinear-LikelihoodMatching , NeuralUCB , Neural-LinUCB , NeuralTS , and NPR . We evaluated all bandit algorithms under the metric of regret/reward and running time, as long as the model sensitivity to the choices of neural netowrk architectures and hyper-parameter settings. We conclude that in most cases, neural contextual bandit algorithms provide significant performance improvement compared to the linear model, while in some specific cases, the advantage of neural bandits is marginal. Besides, the results demonstrate that across different datasets and problem settings, different neural contextual bandit algorithms show various patterns. In other words, no single neural bandit algorithm outperforms others in every bandit problem.

## 2 ALGORITHMS

In this section, we first introduce the general setting of contextual bandit problem, and then present the existing bandit solutions, including both linear and neural models.

### 2.1 CONTEXTUAL BANDIT PROBLEM

We focus on the problem of contextual bandits, where the agent iteratively interacts with the environment for $T$ rounds. $T$ is known beforehand. At each round, the agent will choose one arm from $K$ candidate arms, where each arm is associated with a $d$-dimensional context vector: $\mathbf{x}_a \in \mathbb{R}^d$. Once the arm $a_t$ is selected, the agent will receive the corresponding reward $r_{t,a_t}$ that generated as $r_{t,a_t} = h(\mathbf{x}_{t,a_t}) + \eta_t$, where $h$ is an unknown reward mapping and $\eta_t$ is $\upsilon$-sub-Gaussian noise. The goal of a bandit algorithm is to minimize the pseudo regret:

$$R_T = \mathbb{E}\left[\sum\nolimits_{t=1}^{T} \left(r_{t,a_t^*} - r_{t,a_t}\right)\right], \tag{2.1}$$

where $a_t^*$ is the optimal arm at round t with the maximum expected reward.

### 2.2 LINEAR CONTEXTUAL BANDIT ALGORITHMS

In linear contextual bandits, the unknown reward function $h(\cdot)$ is assumed to be a linear function: $h(\mathbf{x}_{t,a_t}) = \mathbf{x}_{t,a_t}^\top \boldsymbol{\theta}^*$, where $\boldsymbol{\theta}^* \in \mathbb{R}^d$ is the underlying unknown model weight. One of the most popularl linear contextual bandit algorithms is LinUCB (Li et al., 2010; Abbasi-Yadkori et al., 2011). At each round $t$, a ridge regression is applied to learn the current model $\boldsymbol{\theta}_t$ based on the observations collected so far,

$$\boldsymbol{\theta}_t = \arg\min_{\boldsymbol{\theta}} \sum\nolimits_{\tau=1}^{t-1} (r_{\tau,a_\tau} - \mathbf{x}_{\tau,a_\tau}^\top \boldsymbol{\theta})^2 + \frac{\lambda}{2}\|\boldsymbol{\theta}\|_2^2, \tag{2.2}$$

where $\lambda$ is the coefficient of $L_2$ regularization. Then, LinUCB pulls the arm with highest upper confidence bound:

$$a_t = \arg\max_{a \in [K]} \left\{\mathbf{x}_{t,a}^\top \boldsymbol{\theta}_t + \alpha_t \sqrt{\mathbf{x}_{t,a}^\top \mathbf{A}_t^{-1} \mathbf{x}_{t,a}}\right\}, \quad \mathbf{A}_t = \lambda\mathbf{I} + \sum\nolimits_{\tau=1}^{t-1} \mathbf{x}_{\tau,a_\tau} \mathbf{x}_{\tau,a_\tau}^\top \tag{2.3}$$

where $\alpha_t > 0$ is a scaling factor that controls the exploration rate. Once the reward of the pulled arm is received, the model will be updated to $\boldsymbol{\theta}_{t+1}$. By leveraging the width of confidence interval of reward estimation, LinUCB well balances the explore-exploit trade-off in bandit learning and obtains a sublinear regret with respect to the time horizon $T$.

### 2.3 NEURAL BANDIT ALGORITHMS

Numerous attempts have been made to apply neural networks in contextual bandit problems, under the fact that neural networks are remarkable approximators of any unknown functions (Cybenko, 1989). In the following sections, we categorize existing neural contextual bandit algorithms into three main categories based on their exploration methods.

### 2.3.1 NEURAL NETWORK AS FEATURE MAPPING

The first category of algorithms treats neural networks as non-linear feature mappings and deploys linear bandit models on top of the learned mappings. (Riquelme et al., 2018; Zahavy & Mannor, 2019) first introduced a linear exploration policy on the last layer of a neural network. Different from linear bandit algorithms where the feature mapping is stationary, neural-linear algorithms consider the feature mapping changes after the model update at each round.

**NeuralLinear.** The NeuralLinear algorithm (Riquelme et al., 2018) introduced a fully-connected neural network to capture the non-linear relationship between input context vector and the reward. It applies the Bayesian linear regression on the last layer of the neural network (Snoek et al., 2015), and makes the decision via Thompson Sampling. The goal of the neural network is to find a good representation for Bayesian linear regression to predict the reward.

At round $t$, the NeuralLinear algorithm learns the model by minimizing the mean squared error (MSE),

$$\mathcal{L}(\mathbf{w}) = \|f(\mathbf{x}_{a_\tau}; \boldsymbol{\theta}) - r_{\tau, a_\tau}\|_2^2. \tag{2.4}$$

The exploration is performed by using $\boldsymbol{\phi}_t$, the representation learned as the last layer of the neural network. After observing the raw context vector $\mathbf{x}_t$, the agent applies the neural network to learn a representation $\boldsymbol{\phi}_t$. Then $\boldsymbol{\phi}_t$ is used to perform a Bayesian linear regression.

The agent computes the posterior reward of an action via a linear function mapping: $r_t = \boldsymbol{\phi}_t^\top \widehat{\boldsymbol{\mu}}$. After observing $r$, the prior at time $t$ is updated by $Pr(\boldsymbol{\mu}, \boldsymbol{\nu}^2) = Pr(\boldsymbol{\mu}|\boldsymbol{\nu}^2)Pr(\boldsymbol{\nu}^2)$ based on the assumption that $\boldsymbol{\nu}^2 \sim InvGamma(c_t, b_t)$ and $Pr(\boldsymbol{\mu}|\boldsymbol{\nu}^2) \propto \mathcal{N}(\boldsymbol{\mu}_t, \boldsymbol{\nu}^2(\mathbf{A}_0 + \mathbf{A}_t)^{-1})$, where $\mathbf{A}_t$ is defined over the history representations of input data.

Specifically, at each step, we sample the noise parameter $\boldsymbol{\nu}^2$ from $Pr(\boldsymbol{\nu}^2)$ and then sample a weight vector $\widehat{\boldsymbol{\mu}}$ from its posterior distribution $\mathcal{N}(\boldsymbol{\mu}_t, \boldsymbol{\nu}^2(\mathbf{A}_0 + \mathbf{A}_t)^{-1})$. With the sampled $\widehat{\boldsymbol{\mu}}$, we select the arm by $a_t = \arg\max_{a \in [K]} \boldsymbol{\phi}_t^\top \widehat{\boldsymbol{\mu}}$ and then observe the reward $r_t$. The parameters in NeuralLinear are calculated as follows:

$$\mathbf{A}_t = (\boldsymbol{\Phi}^\top \boldsymbol{\Phi} + \boldsymbol{\Lambda}_0)^{-1}, \qquad \boldsymbol{\mu}_t = \mathbf{A}_t(\boldsymbol{\Lambda}_0 \boldsymbol{\mu}_0 + \boldsymbol{\Phi}^\top \boldsymbol{R}), \tag{2.5}$$

$$c_t = c_0 + t/2, \qquad b_t = b_0 + (\boldsymbol{R}^\top \boldsymbol{R} + \boldsymbol{\theta}_0^\top \mathbf{A}_0 \boldsymbol{\theta}_0 - \boldsymbol{\theta}_t^\top \mathbf{A}_t^{-1} \boldsymbol{\theta}_t)/2, \tag{2.6}$$

where $\boldsymbol{\mu}_0 = 0$, $\boldsymbol{\Lambda}_0 = \lambda I_d$, and $\mathbf{A}$ is a matrix defined based on the history representation of input data. In addition, $\boldsymbol{\Phi}$ and $\boldsymbol{R}$ can be viewed as memory buffers storing the history representation data and reward, respectively.

**NeuralLinear with Likelihood Matching.** This algorithm (Zahavy & Mannor, 2019) extends NeuralLinear with a small memory buffer to handle the catastrophic forgetting problem, which refers to the issue of drifting model estimation caused by the loss of information from previous experience (Kirkpatrick et al., 2017). At each round $t$, it stores the representation into a bounded memory buffer, which is denoted as $E$. When $E$ is full, it will remove a previous observation in a round robin manner.

The likelihood matching mechanism is to deal with the change of representation by using the DNN and the memory buffer. Based on the posterior distribution of $\boldsymbol{\theta}_t \sim \mathcal{N}(\boldsymbol{\theta}_t, \boldsymbol{\nu}^2(\mathbf{A}_0 + \mathbf{A}_t)^{-1})$, the marginal distribution of $r_t$ is $\mathcal{N}(\boldsymbol{\phi}_t^\top \boldsymbol{\theta}_t, \boldsymbol{\nu}^2 s_t^2)$, where $s_t = \sqrt{\boldsymbol{\phi}_t^\top \mathbf{A}_t^{-1} \boldsymbol{\phi}_t}$ (Agrawal & Goyal, 2013). Thus the goal is to make the likelihood of $r_t$ given the new feature mappings consistent with it given the old feature mappings.

After each training phase, new feature representation is denoted as $E_\phi \in \mathbb{R}^{n \times m}$, where $n$ is the length of the previous action sequence and $m$ is the dimension of the feature representation. Use $\mathbb{E}_{\phi_{old}}$ to denote the old representation before training. The likelihood matching approach summarizes the old representation into the priors of the correlation matrix $\mathbf{A}^0$ and the mean vector $\boldsymbol{\mu}^0$ under the new representation. The weights of the last layer of the neural network $\boldsymbol{\mu}$ is a good approximation of the mean $\boldsymbol{\mu}^0$ because the neural network is trained online by holding the information of the entire observed data and therefore not limited to the memory buffer. For the approximation of the correlation matrix $\mathbf{A}^0$, the goal is to find $\mathbf{A}^0$ such that

$$s_t^2 = \boldsymbol{\phi}_t^\top (\mathbf{A}^0)^{-1} \boldsymbol{\phi}_t = \text{Trace}((\mathbf{A}^0)^{-1} \boldsymbol{\phi}_t^\top \boldsymbol{\phi}_t),$$

where $s_t^2 = \boldsymbol{\phi}_{old}^\top (\mathbf{A}_{old})^{-1} \boldsymbol{\phi}_{old}$ and the equality is based on the cyclic property of the trace. With the definition that $Z_t = \boldsymbol{\phi}_t^\top \boldsymbol{\phi}_t$, the problem can be viewed as a regression problem:

$$\underset{(\mathbf{A}^0)^{-1}}{\text{minimize}} \sum_{j=1}^n (\text{Trace}((Z_j^\top \mathbf{A}^0)^{-1}) - s_j^2)^2, \quad \text{subject to } (\mathbf{A}^0)^{-1} \geq 0.$$

The exploration step is similar to the NeuralLinear algorithm:

$$\begin{aligned}
\mathbf{A}_t &= \mathbf{A}_{t-1} + \boldsymbol{\phi}_t^\top \boldsymbol{\phi}_t, & \boldsymbol{\Psi}_t &= \boldsymbol{\Psi}_{t-1} + \boldsymbol{\phi}_t^\top r_t, \\
R_t^2 &= R_{t-1}^2 + r_t^2, & \boldsymbol{\theta}_t &= (\mathbf{A}^0 + \mathbf{A}_t)^{-1}(\mathbf{A}^0 \boldsymbol{\mu}^0 + \boldsymbol{\Psi}_t), \\
c_t &= c_0 + t/2, & b_t &= b_0 + (R_t^2 + (\boldsymbol{\mu}^0)^\top \mathbf{A}^0 \boldsymbol{\mu}^0 - \boldsymbol{\theta}_t^\top \mathbf{A}_t \boldsymbol{\theta}_t)/2,
\end{aligned}$$

where $\boldsymbol{\theta}_t$ is a weight vector sampled from the posterior distribution, $\boldsymbol{\mu}^0$ is the mean prior, and $\mathbf{A}^0$ is the prior of the correlation matrix.

**NeuralLinUCB.** A neural network always have the number of parameters in the order of 100 thousands, which makes the exploration on the entire parameter space inefficient. The NeuralLinUCB is a combination of the NeuralLinear and the NeuralUCB algorithm. It introduces a neural network to learn a deep representation and then performs UCB-based exploration on the last layer of the neural network. In particular, the reward function is defined as the inner product between the weight of the last layer of the neural network and the last hidden layer representation, namely, $\boldsymbol{r} = \boldsymbol{\phi}^\top \boldsymbol{\mu}_{t-1}$. Then a UCB-based exploration is performed as follows:

$$a_t = \underset{a \in [K]}{\arg\max} \left\{ \boldsymbol{\phi}_t^\top \boldsymbol{\theta}_{t-1} + \alpha_t \sqrt{\boldsymbol{\phi}_t^\top \mathbf{A}_{t-1}^{-1} \boldsymbol{\phi}_t} \right\}, \tag{2.7}$$

where $\boldsymbol{\theta}_{t-1}$ is a point estimator of the unknown weight in the last layer, $\boldsymbol{\phi}_t$ is the representation learned as the last layer of the neural network and $\mathbf{A}_t$ is a matrix defined based on the history representation of input data.

### 2.3.2 NEURAL TANGENT KERNEL BASED ALGORITHMS

Most recently, under the neural tangent kernel space, neural bandit algorithms are able to perform the exploration in the entire parameter space. In this category, a fully connected neural network $f(\cdot)$ is introduced to approximate the reward $h(\boldsymbol{x})$,

$$f(\mathbf{x}; \boldsymbol{\theta}) = \sqrt{m} \mathbf{W}_L \sigma(\mathbf{W}_{L-1} \sigma(\dots \sigma(\mathbf{W}_1 \mathbf{x}))), \tag{2.8}$$

where $\sigma(x) = \text{ReLU}(x)$, $\boldsymbol{\theta} = [\text{vec}(\mathbf{W}_1), \dots, \text{vec}(\mathbf{W}_L)] \in \mathbb{R}^p$, with $p$ as the number of parameters of all hidden layers of the neural network, and $p = m + md + m^2(L-1)$ with $m$ as the width of the each hidden layer.

**NeuralUCB.** At round $t$, NeuralUCB learns the model by minimizing an $l_2$-regularized square loss,

$$\mathcal{L}(\boldsymbol{\theta}) = \sum_{\tau=1}^t \left( f(\mathbf{x}_{a_\tau}; \boldsymbol{\theta}) - r_{\tau,a_\tau} \right)^2 / 2 + m\lambda \|\boldsymbol{\theta} - \boldsymbol{\theta}_0\|_2^2 / 2. \tag{2.9}$$

where the regularization centers at the randomly initialization $\boldsymbol{\theta}_0$ with the trade-off parameter $\lambda$. In NeuralUCB, with neural tangent kernel, it is proved that with a satisfied neural network width $m$, with high probability, the underlying reward mapping function can be approximated by a linear function over $\mathbf{g}(\mathbf{x}; \boldsymbol{\theta}_0)$, parameterized by $\boldsymbol{\theta}^* - \boldsymbol{\theta}_0$, where $\mathbf{g}(\mathbf{x}; \boldsymbol{\theta}_0) = \nabla_{\boldsymbol{\theta}} f(\mathbf{x}; \boldsymbol{\theta}_0) \in \mathbb{R}^d$ is the gradient of the initial neural network. Therefore, at each round, NeuralUCB selects the arm as,

$$a_t = \underset{a \in [K]}{\arg\max} \left\{ f(\mathbf{x}_{t,a}; \boldsymbol{\theta}_{t-1}) + \alpha_t \sqrt{\mathbf{g}(\mathbf{x}_{t,a}; \boldsymbol{\theta}_{t-1})^\top \mathbf{A}_t^{-1} \mathbf{g}(\mathbf{x}_{t,a}; \boldsymbol{\theta}_{t-1})} \right\}, \tag{2.10}$$

where $\alpha_t$ is a positive scaling factor, $\boldsymbol{\theta}_{t-1}$ is the current parameter of neural network, and $\mathbf{A}_t$ is a matrix defined based on history gradient of the neural network,

$$\mathbf{A}_t = \sum_{\tau=1}^{t-1} \mathbf{g}(\mathbf{x}_{\tau,a_\tau}; \boldsymbol{\theta}_0) \mathbf{g}(\mathbf{x}_{\tau,a_\tau}; \boldsymbol{\theta}_0)^\top / m + \lambda \mathbf{I}. \tag{2.11}$$

**NeuralTS.** The NeuralTS algorithm is similar to the design of NeuralUCB based on the neural tangent kernel technique (Jacot et al., 2018). Similar to NeuralUCB, NeuralTS learns the model

Table 1: Statistics of UCI dataset

| DATASET | Mushroom | Covertype | Magic | Adult | Shuttle |
|---|---|---|---|---|---|
| Number of attributes | 22 | 54 | 11 | 14 | 9 |
| Number of arms | 2 | 7 | 2 | 2 | 7 |
| Number of instances | 8,124 | 581,012 | 19,020 | 48,842 | 58,000 |

parameters by minimizing Eq equation 2.9. NeuralTS explores the neural network parameter space via Thompson Sampling, where it maintains a posterior distribution of the reward estimation for each arm. At each round $t$, for each arm, NeuralTS samples the reward from its posterior distribution,

$$r_{t,a_t} \sim \mathcal{N}(f(\mathbf{x}_{t,a_t}; \boldsymbol{\theta}_{t-1}), \nu^2 \sigma^2), \tag{2.12}$$

where $\nu$ is the exploration variance parameter, and $\sigma_t^2$ is computed as:

$$\sigma_t^2 = \lambda \mathbf{g}(\mathbf{x}_{t,a_t}; \boldsymbol{\theta}_{t-1})^\top \mathbf{A}_t^{-1} \mathbf{g}(\mathbf{x}_{t,a_t}; \boldsymbol{\theta}_{t-1}),$$

with $\mathbf{A}_t$ defined in Eq 2.11.

### 2.3.3 PERTURBATION BASED ALGORITHMS

The third category of methods avoids explicit exploration by introducing controlled perturbations in the neural network. (Jia et al., 2021) introduced pseudo noise generated from a zero-mean Gaussian distribution to the observed reward history, which eliminated explicit exploration in neural bandit algorithms.

**NPR.** At each round $t$, NPR updates the model by minimizing the loss function defined as,

$$\mathcal{L}(\boldsymbol{\theta}) = \sum_{\tau=1}^{t} \left( f(\mathbf{x}_{\tau,a_\tau}; \boldsymbol{\theta}) - (r_{\tau,a_\tau} + \gamma_\tau) \right)^2 / 2 + m\lambda \|\boldsymbol{\theta} - \boldsymbol{\theta}_0\|_2^2 / 2. \tag{2.13}$$

where $\{\gamma_s^t\}_{s=1}^t \sim \mathcal{N}(0, \sigma^2)$ are Gaussian random variables that are independently sampled in each round $t$, and $\sigma$ is a hyper-parameter that controls the strength of perturbation and thus the exploration. Because of the perturbation, the agent only need to select the arm with the largest estimated reward:

$$a_t = \arg\max_{a \in [K]} f(\mathbf{x}_a; \boldsymbol{\theta}_{t-1}) \tag{2.14}$$

where $f(\mathbf{x}; \boldsymbol{\theta}_{t-1})$ is the output of the neural network. NPR is proved to obtain the same level of regret upper bound as other neural bandit algorithms.

## 3 EXPERIMENTS

In this section, we present the empirical evaluations of all the neural contextual bandit algorithms introduced in Section2. In particular, we report the results on five K-class classification datasets from UCI machine learning datasets (Dua & Graff, 2017), a learning to rank dataset for web search: Web10K (Qin & Liu, 2013), and two logged bandit datasets for recommendations: Japanese Fashion datasets (Saito et al., 2020), and Yahoo! Front Page Module dataset(Li et al., 2010).

### 3.1 DATASET AND EXPERIMENT SETTINGS

#### 3.1.1 K-CLASS CLASSIFICATION DATASETS

We evaluate all neural bandit algorithms on five datasets from UCI machine learning repository. Specifically, we use datasets `mushroom`, `covertype`, `shuttle`, `adult`, and `magic`. These are K-class classification datasets, of which the statistics are presented in Table 1. We adopt the disjoint model Li et al. (2010) to build the context feature vectors to generated a K-armed pool: $\mathbf{x}_1 = (\mathbf{x}, \mathbf{0}, \dots, \mathbf{0}), \dots, \mathbf{x}_k = (\mathbf{0}, \dots, \mathbf{0}, \mathbf{x}) \in \mathbb{R}^{d \times k}$. The agent receives reward 1 if the correct class is selected, otherwise 0. Cumulative regret is defined as the total mistakes made by the agent over $T$ rounds. We report the averaged cumulative regret across 10 runs for 10,000 rounds, except for the `mushroom` dataset which only contain 8,124 instances in Figure 1(a) to 1(e). For neural bandit algorithms, we apply a 3-layer neural network with $m = 16$ units in each hidden layer and the model is updated every round.

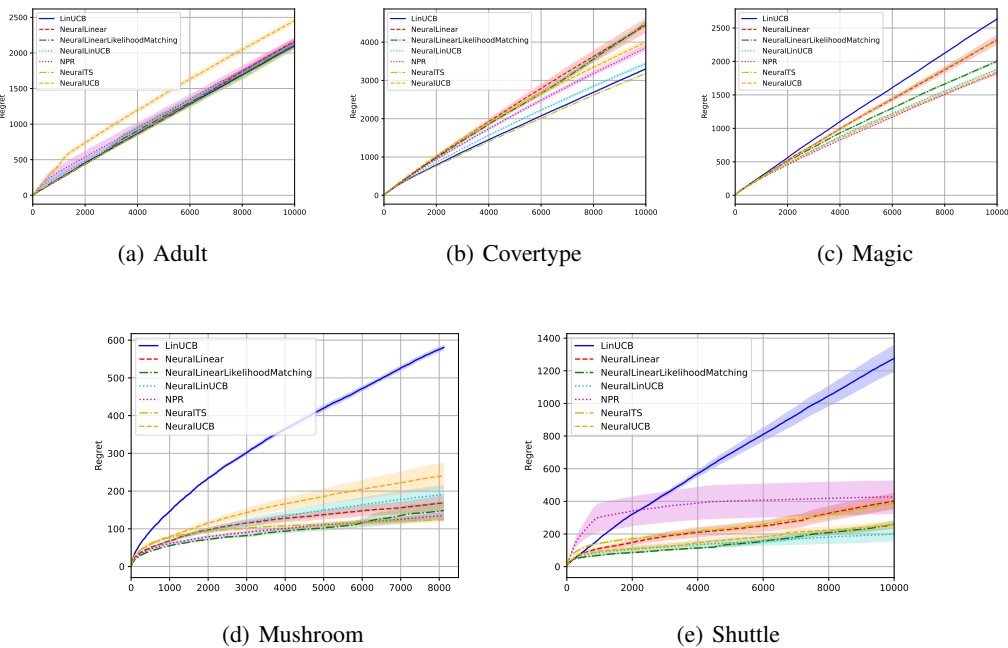

Figure 1: Empirical results of regret and time consumption on UCI dataset.

### 3.1.2 LEARNING TO RANK DATASET

The MSLR-Web10K dataset is a public learning to rank dataset from Microsoft LETOR benchmark, contains 10,000 unique queries, each containing a set of documents. Each document is associated with 136 ranking features and labeled with a relevance degree range from not relevant(0) to perfectly relevant(4). We treat the documents in each query as arms. At each round, we randomly select one query and treat the corresponding documents as the arm pool. The regret is defined as the difference between the best relevance score of the arm pool and the selected one. In our experiment, we applied a 3-layer neural network with $m = 64$ units in each hidden layer. Figure 3 shows the averaged regret across 5 runs for 150,000 rounds.

### 3.1.3 JAPANESE FASHION DATASETS

The Japanese Fashion datasets is a set of logged bandit datasets collected from a e-commerce platform, ZOZOTOWN. The dataset was collected in a 7-day experiment by using two different policies: `random` and `bernoulli thompson sampling`. It includes three "campaigns", corresponding to "ALL", "Men's" and "Women's" items, respectively. In our simulation, we only use the `random` collected dataset with "ALL" items, which contains 1,374,237 user-item interactions and 80 items. Each item has 4 features and each user is represented with a 26-dimension binary feature vector. We generated the candidate pool as follows: we fixed the size of the candidate arm pool to $k = 25$ for each round; for each user, we selected the item according to the complete observations in the dataset, and randomly choose 24 items from the item list. We generated the context vectors by computing the outer product of user feature and item feature. In our experiment, we adopted a 3-layer neural network with $m = 32$ units in each hidden layer. Cumulative CTR is used to compare the performance of different algorithms, which is defined as the number of clicks it obtains and the number of accesses it is received. To improve visibility, we normalized the cumulative CTR by a random strategy's cumulative CTR, which is the algorithm's `relative` CTR (Li et al., 2010). We ran through the dataset 5 times and reported the averaged relative CTR in Figure 4.

### 3.1.4 YAHOO! FRONT PAGE TODAY MODULE DATASET

The Yahoo! Front Page Module dataset is collected in May 2009. In each observation, users were randomly selected to visit a small set of articles hand-picked from a large article pool, where old

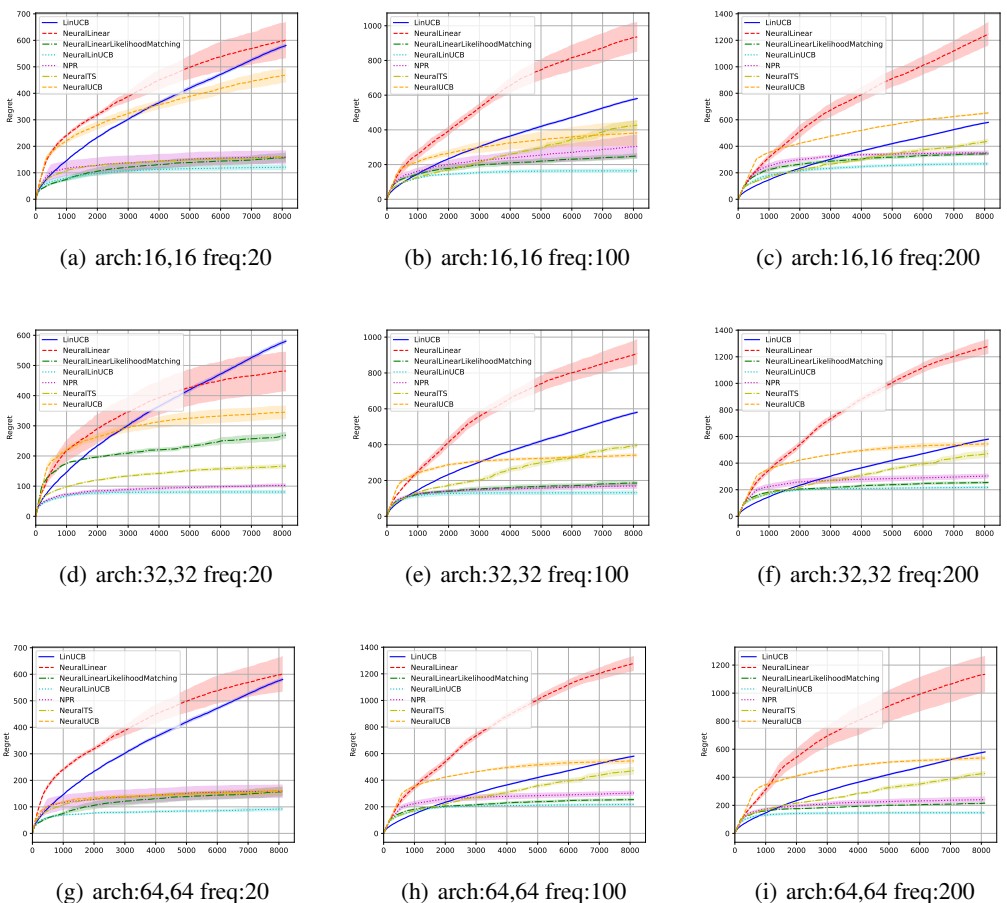

Figure 2: Sensitivity evaluation on the mushroom dataset.

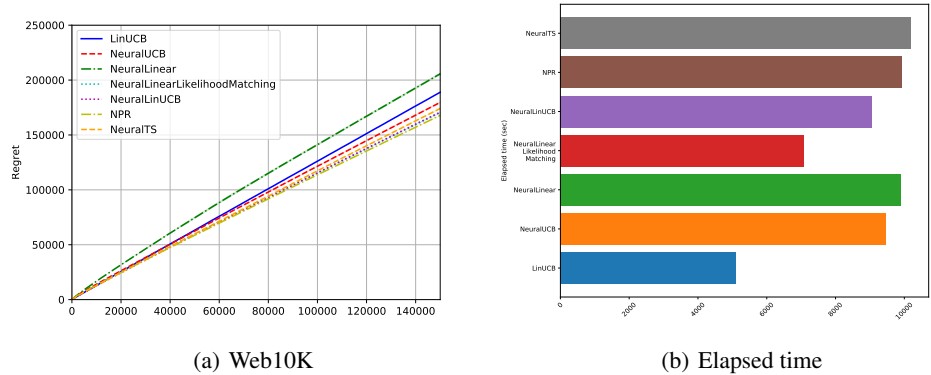

(a) Web10K

(b) Elapsed time

Figure 3: Cumulative regret and time consumption on Web10K dataset.

articles will be replaced by the new ones after a period of time. The size of the candidate pool is 20 on average.

We treat the clicked articles for each user as positive feedback. We constructed the context vector by computing the outer product of user feature and article feature, the concatenating the outer product with the user feature and article feature. In our experiment, we select May01 dataset, which contains more than 4.7 million events. A 3-layer neural network with $m = 16$ units in each hidden layer was applied. Following the evaluation metric in Japanese Fashion dataset, we report the averaged Relative CTR across 5 runs in Figure 5.

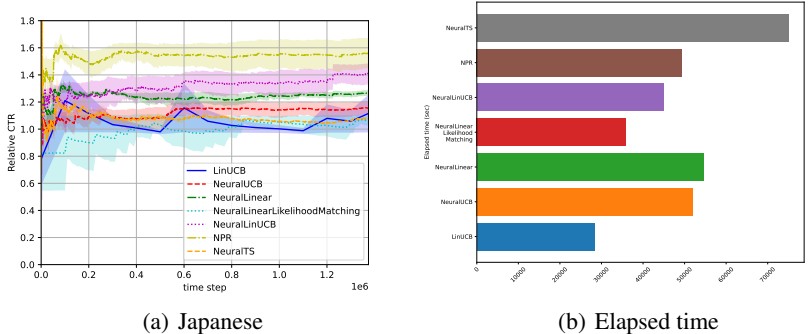

(a) Japanese          (b) Elapsed time

Figure 4: Comparisons of relative CTR and time consumption on Japanese Fashion dataset.

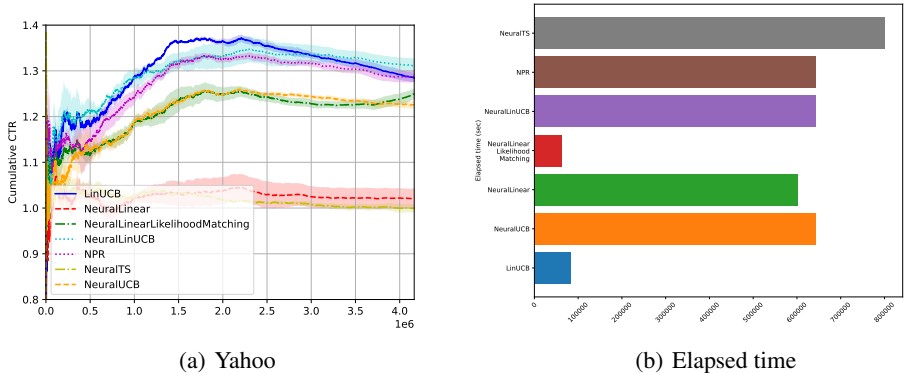

(a) Yahoo          (b) Elapsed time

Figure 5: Cumulative relative CTR and time consumption on Yahoo dataset.

## 3.2 EXPERIMENT RESULTS

### 3.2.1 RESULTS OF K-CLASS CLASSIFICATION

Figure 1 show the cumulative regret on K-class classification datasets. It can be observed that LinUCB fails as it cannot capture the nonlinear relationship between the context vector and the reward. In contrast, thanks to the power of representation learning of neural networks, the performance is strongly boosted by neural models on `mushroom` and `shuttle` datasets. However, the improvement on `adult`, `covertype` and `magic` is limited.

Although neural bandit algorithms show better or at least comparable performance to the linear bandit algorithm, the variance of neural bandit models is much higher in Figure 1(d) and 1(e), which might be harmful in real-world applications. To further investigate the sensitivity of the neural bandit models, we evaluate three different neural architectures with $m = 16, 32$ and $64$ units in each hidden layer, and three model updating frequencies: $\{20, 100, 200\}$. Due to space limit, we report the result on the `mushroom` dataset in Figure 2 and leave the other four `covertype`, `shuttle`, `magic`, and `adult` in the appendix.

The performance of neural bandit models depends on their converge speed. Reducing updating frequency slow down the convergence of neural bandit algorithms. The NeuralLinear algorithm even fails to converge in most of the nine settings. Infrequently updating model parameters is not helpful for experiments on small datasets like these K-class classification datasets. Based on the results, we conclude that increasing the width of the neural network helps models to converge and reduce the variance since larger neural networks can capture more information, and increasing the update frequency in the initial steps will speed up the convergence.

### 3.2.2 Results on Web10K dataset

Figure 3 shows the averaged cumulative regret and time consumption of finishing 15,000 rounds on the Web10K dataset. The neural bandit algorithms, except for NeuralLinear, consistently outperform the linear bandit algorithm. However, the advantage of applying neural bandit models is not apparent, and the LinUCB algorithm uses almost half of the time less than most neural bandit algorithms. Among the neural bandit algorithms, the NeuralLinear with LikelihoodMatching algorithm shows promising results with the least running time. The limited memory buffer seems efficient and can capture most of the valuable information of the historical data when running a large-scale experiment.

### 3.2.3 Results on Japanese Fashion Dataset

Figure 4 shows the averaged cumulative relative CTR and time consumption on the Japanese Fashion dataset. The performance is boosted by neural models. The neural bandit algorithms strongly outperform the linear bandit algorithm. Compared with the Web10K dataset, the Japanese Fashion dataset contains more information. It has two sides of features: the user feature and the item feature. The interaction of the user and item feature provides informative knowledge to the neural network, which helps it to capture more detailed information from the data. For time consumption, the NeuralLinear with LikelihoodMatching uses the least time to finish the experiment. At the same time, the LinUCB algorithm spends less time than all of the neural bandit algorithms. The NPR algorithm achieves the highest cumulative relative CTR, and the NeuralLinUCB also performs well.

### 3.2.4 Results on Yahoo dataset

Figure 5 provides the averaged cumulative relative CTR and time consumption on the Yahoo dataset. The linear bandit algorithm outperforms most neural bandit algorithms except the NeuralLinUCB algorithm. As the tiny document and user features, we conclude that simple features might fail to provide helpful knowledge for training a neural model. The linear model is good enough to capture low-level information from a simple feature. We surprisingly found that the NeuralLinear with LikelihoodMatching algorithm used the least time among all of the algorithms, even faster than the linear bandit algorithm, while the performance is not bad. It only used less than twenty percent of the time other neural bandit algorithms used, which is a strong advantage in such a large-scale dataset.

## 4 Conclusion

In this work, we provide an inclusive empirical study to investigate the impact on the performance of applying neural networks in contextual bandit algorithms. We found that the neural bandit algorithms can capture more nonlinear information and show promising results in most cases. The neural bandit algorithms might fail if the number of data is insufficient, like the datasets from UCI machine learning, or the context feature is too simple to provide enough knowledge to learn. The NeuralLinUCB and the NPR algorithm always perform the best among all neural bandit algorithms. In contrast, the NeuralLinear with LikelihoodMatching algorithm is the only one that can leverage the performance and time consumption. Some neural bandit algorithms would prefer frequent model updating in the beginning of the experiment. It is worth investigating further and developing the neural bandit algorithms.

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
