# OpenReview forum: "An Empirical Study of the Neural Contextual Bandit Algorithms"
_ICLR.cc/2023/Conference — Submitted to ICLR 2023_

### Official Review · Reviewer_22TM · 2022-10-19

**Confidence:** 4
**Correctness:** 2
**Technical Novelty And Significance:** 1
**Empirical Novelty And Significance:** 1
**Recommendation:** 3

**Clarity, Quality, Novelty And Reproducibility:**

While the authors spend a lot of time to describe the literature, the experiments are insufficiently described.

In Learning to rank, Japanese fashion and Yahoo datasets, the number of arms is not provided to the reader. Moreover, some of the tested problems seem particular. For instance variable number of arms in Learning to rank or Japanese fashion (the reviewer does not understand at all this experiment), combinatorial number of arms in Yahoo dataset (20 arms chosen from ?).
The authors should accurately describe each experiment, and how they have adapted each algorithm to each problem. For instance UCI datasets corresponds to contextual bandits with the observation of a single context at each time step, while Japanese Fashion dataset look likes to a structured bandit problem (a context per arm), while Yahoo dataset seems to be an hybrid problem (a context per arm and a general context at each time step).
In figure 5, why is cumulative relative CTR decreasing?
Why is the number of hidden units not the same for all experiments? How are they chosen?
Finally, it lacks a metric (for instance ranking of algorithms) and a statistical test to sum up the results.


**Strength And Weaknesses:**

The empirical evaluation is made on several datasets. However, the authors spend 5 pages for describing the state-of-the-art, and that is a lot for a nine pages paper. Moreover, some of the described algorithms have pseudo-regret upper bounds, but they are not given. In the case of bandit algorithms, it is however a strong argument since they are often used in “cold start”.
Finally, despite the fact that they are nonlinear contextual bandit algorithms, and some of them are cited (Kernel UCB, Generalized linear bandit), they are not tested. It is a pity, knowing that the main conclusion of the paper is that neural contextual bandit algorithms can capture nonlinear information.

Some missed references:
Efficient Bandit Algorithms for Online Multiclass Prediction, ICML 2008 (Banditron is the first attempt to use neural networks in contextual bandit problem)
Efficient optimal learning for contextual bandits, UAI 2011 (oracle based contextual bandit)
A Neural Networks Committee for the Contextual Bandit Problem, ICONIP 2014 (first attempt to use neural networks in contextual bandit problem that practically works).
Random Forest for the Contextual Bandit Problem, AISTATS 2016 (contextual bandit based on a non-linear model).
Taming the Monster: A Fast and Simple Algorithm for Contextual Bandits, ICML 2014 (oracle based contextual bandit)

Minor comment:
Deep neural networks are not recent (page 1).


**Summary Of The Paper:**

This paper proposes to investigate the empirical performances of neural contextual bandit algorithms. LinUCB is used as a baseline. The algorithms are tested on four type of problems: UCI classification dataset, MSLR-Web10K dataset, Japanese fashion dataset, and Yahoo front page dataset.
The conclusion of the paper is that in comparison to LinUCB, neural contextual bandit algorithms can capture nonlinear information.


**Summary Of The Review:**

Rather than spending 5 pages to describe state-of-the-art, the author should accurately describe their experiments.
________________________________________________________________________________________________________
I thank the authors for their answers.
I raised my score, but I do not vote for acceptance.

---

> ### Author Response · Authors · 2022-11-18
> **Response to Reviewer 22TM**
>
> We sincerely thank the reviewer for the comments and suggestions. Please find our responses in the following.
>
> 1. **"Some missed references: Efficient Bandit Algorithms for Online Multiclass Prediction, ICML 2008 (Banditron is the first attempt to use neural networks in contextual bandit problem) ……"**
>
> Thanks for the suggestions! We have included these papers in our paper now.
>
>
> 2. **"For instance variable number of arms in Learning to rank or Japanese fashion (the reviewer does not understand this experiment at all), combinatorial number of arms in Yahoo dataset (20 arms chosen from ?)."**
>
> The Japanese dataset is a set of logged datasets collected by a fashion e-commerce platform of Japan. This platform uses multi-armed bandit algorithms to recommend items. The data was collected in a 7-day experiment on three “campaigns,” corresponding to “ALL”, “Men’s” and “Women’s” items, respectively. Each “campaign” uses random policy and Bernoulli TS policy for each user impression, respectively. For each user, the policy selects three items to display. This dataset assumes that the reward only depends on the item and its position, while in our experiment, we assume the reward depends only on the item and user. The feature vectors are hashed due to the privacy issue. This is a large scale dataset and contains millions of user-item interactions. In our experiment, we only use the dataset collected by random policy on campaign “ALL”. “ALL” contains 80 items, each of which is associated with a 4-dimensional feature vector.  Each user is associated with a 26-dimensional binary feature vector and each item is associated with a 4-dimensional feature vector. Each row of data represents a user-item interaction, including the user iD, item that was chosen to display to this user, the position of the item, click or not, and past click histories of the user.
> We fixed the size of the candidate arm pool to k = 25 for each round; for each user, we choose the item displayed to the user according to the observation, and then randomly choose the other 24 items from “ALL” items.  The click is treated as the reward. We computed the outer product of the user and item feature as the input, which is a 104-dimensional feature vector, of the model. We report an algorithm’s relative CTR, which is the algorithm’s CTR divided by the random’s CTR as it is always 1 by definition.
>
>
> 3. **"In figure 5, why is cumulative relative CTR decreasing? "**
>
> One reason might be because the candidate arm provided by the yahoo dataset will change over time.The model needs more time to capture the information of new arms. Also, the result reported in the paper is based on 5 separate runs.We will do further experiments to check if the model can learn useful information in multiple runs.
>
> 4. **"Why is the number of hidden units not the same for all experiments? How are they chosen?"**
>
> We consider different neural architectures based on the complexity of feature space and efficiency. In Yahoo Dataset, each user is associated with a 5-dimension feature vector and each item is associated with a  6-dimension feature vector. We first compute the outer product of item and user feature and get a 30-dimension vector. Then we concatenate it with the raw user and item feature vector as the input context feature vector of the model.  The length of the constructed context feature vector of Yahoo is 41. For the Japanese dataset, we have a 26-dimensional user feature vector and 4-dimensional item feature vector. We computed the outer product of the user and item feature as the input context feature vector of the model. The Web10K dataset provides a 136-dimensional feature vector for each document. The length of the constructed feature vector  of the Japanese and  Web10K dataset is 104, 136 respectively. We choose 16 * 16 architecture for the Yahoo dataset, 32 * 32 for Japanese Fashion dataset and 64 * 64 for the Web10K dataset based on the complexity of their constructed context feature vector. In practice, as the Yahoo dataset is a large scale dataset that contains more than 4 million instances, a complex model is computationally expensive in this case.

---

### Official Review · Reviewer_Zjj4 · 2022-10-20

**Confidence:** 4
**Clarity, Quality, Novelty And Reproducibility:** not reproducible. no code.
**Correctness:** 3
**Technical Novelty And Significance:** 1
**Empirical Novelty And Significance:** Not applicable
**Recommendation:** 3

**Strength And Weaknesses:**

I feel this paper is better suitable for a workshop paper rather than a full conference paper. This paper didn't provide any new algorithm / theory / dataset. The large body of approximate Thompson sampling algorithms ("Deep Bayesian Bandits Showdown: An Empirical Comparison of Bayesian Deep Networks for Thompson Sampling") are not compared.

The main criticism of neural contextual bandits algorithm is the mismatch between algorithm and theory. I feel experiments should carefully check if the assumption of NTK is not satisfied, will the algorithm still work?

**Summary Of The Paper:**

This paper proposed a empirical evaluation of neural contextual bandits algorithms.

**Summary Of The Review:**

An nice empirical evaluation but no new message.

---

> ### Author Response · Authors · 2022-11-18
> **Response to Reviewer Zjj4**
>
> We sincerely thank the reviewer for the comments and suggestions.
>
> In this work, our main purpose is to perform a horizontal comparison among neural contextual bandit algorithms to inform the community about the empirical performance of different neural contextual bandit algorithms under different circumstances. To this end, theoretical comparison exceeds the scope of this paper.
>
> We agree that the mismatch between theory and practice is the main criticism of neural contextual bandits. In  our results, feature mapping based neural bandit algorithms like NeuralLinUCB always perform better than NTK-based neural bandit algorithms, such as NeuralTS and NeuralUCB. Based on our understanding, the assumption of NTK theory is too strong and it is not practical. In real-world applications, we cannot apply such an extremely wide neural network, which is computationally unaffordable. We always cannot meet the assumption that NTK theory, while feature mapping based neural bandit algorithms don’t have such a strong assumption. That might cause the gap between theory and algorithm in NTK-based neural bandit algorithms.

---

### Official Review · Reviewer_sUDB · 2022-10-22

**Confidence:** 4
**Correctness:** 2
**Technical Novelty And Significance:** 3
**Empirical Novelty And Significance:** 3
**Recommendation:** 3

**Clarity, Quality, Novelty And Reproducibility:**

Quality issues described above.

The exposition is clear.  If you scale up the number of datasets, you will have to elide details to appendices.  Furthermore you will have to take a uniform approach to dataset processing.

For reproducibility, you should provide a link to a public repository with scripts that, when invoked, do *everything*: 1) downloading data sets, 2) downloading code dependencies, etc. leading to 3) producing any graph that appears in the paper.  Empirical evidence is conditional and therefore transparent reproducibility is priority numero uno.

**Strength And Weaknesses:**

Strength:
  * This type of empirical investigation is important and undersupplied in the literature.

Weakness:
  * Empirical evidence is conditional, so to draw broad conclusions, a massive number of datasets need to be used, e.g. Bietti et al [1] consider more than 500.

> The neural bandit algorithms might fail if the number of data is insufficient, like the datasets from UCI machine learning, or the context feature is too simple to provide enough knowledge to learn.

The beautiful thing about using many datasets is you can do meta-data analysis, so instead of just saying something like this, you can quantify it (e.g., what properties of datasets allow one to predict which algo is best?)

[1] https://arxiv.org/abs/1802.04064

**Summary Of The Paper:**

Authors compare multiple neural contextual bandit algorithms on multiple data sets along with non-neural baselines.

**Summary Of The Review:**

Quite simply, if you can increase the number of datasets used to at least 100, I would accept.  Also great would be to quantify claims about when algo FOO is best with meta-data analysis.

If the response is something like:
  * it would take too long to run the algorithms on a large number of datasets ... well ... that's important to know!
  * there's no single architecture that works well on all the datasets ... well ... that's important to know!

I strongly encourage the authors to continue this line of work, as previously stated, it is undersupplied in the literature.  However, you need to up your game: empirical papers require more comprehensive effort.

---

> ### Author Response · Authors · 2022-11-18
> **Response to Reviewer sUDB**
>
> We sincerely thank the reviewer for the comments and suggestions.
>
> To the best of our knowledge, there is no existing work performing such extensive empirical comparisons among the existing neural bandit algorithms. In order to provide an even more comprehensive analysis of the results, we will try to compare the difference of datasets we used in our experiments to quantify the performance of different algorithms under different circumstances.

---

### Official Review · Reviewer_qgBu · 2022-10-24

**Confidence:** 4
**Correctness:** 3
**Technical Novelty And Significance:** 1
**Empirical Novelty And Significance:** 2
**Recommendation:** 3

**Clarity, Quality, Novelty And Reproducibility:**

Clarity:

-[pro] I found the Section 1 and 2 of the paper is easy to read and follow. I do like the overview of the methods.
-[con] The experiments section needs to re-organized. Instead of structuring based on the datasets, it would be beneficial to re-organized based on the problem instance, or the structure of the environment.

Quality/Novelty:

The paper does put some efforts in performing various experiments, but I feel it does not do a good job in analyzing and summarizing the results, which make the paper seems have very few takeaways, and little novelty.

Reproducibility:

The variance is reported, while the code is not attached.

**Strength And Weaknesses:**

Strength:

- Neural Bandits are popular bandit algorithms these days, with the representation power from deep models. These paper studies an important aspect, i.e., how different algorithms perform under various environments, which should be relevant to the community.

- The paper is easy to read and follow.

- I like the Section 2.3, which seems to provide a nice overview/summary of different neural bandit algorithms.

Weakness:

- I appreciate the authors' efforts on performing the experiments on various datasets, to verify the pros and cons of different algorithms. However, at least for me, Section 3 is not informative. The current structure seems like the following: introduce different datasets first, followed by the setup, finally summarized the results from the figures. The results seem only contain some facts without any analysis/reasoning about why certain algorithms perform well under this setting, even say hypothesis. It then would be very hard for the readers to learn from this paper.

- There are actually lots of interesting aspects from the figures, but are not pointed out/analyzed by the authors. For example, it is kind of well-known TS performs better than UCB empirically. However, from Fig.2 and Fig.3, it seems not the case if we use the representation learned by neural networks as the contextual features. It would be great if the authors could digest and summarize the results in a deeper way, that would also increase the impact of the paper.

- Though the number of datasets seems sufficient, I feel the scenarios it covers is limited. For example, it would be interesting to see the case, when the number of arms is large, this is pretty relevant for practical scenarios such as recommender systems.

**Summary Of The Paper:**

This paper provides an empirical study of the neural bandit algorithms, which are the type of bandit algorithms that combine the benefit of deep neural networks, along with the simplicity of the classical linear bandit algorithms. It chooses the linear model: LinUCB as the reference algorithm, and studies how the recently proposed neural bandit models, i.e., NeuralLinear, NeuralLinear-LikelihoodMatching, NeuralUCB , Neural-LinUCB, NeuralTS, and NPR compares with the reference, under different datasets.


**Summary Of The Review:**

This paper studies an important topic, i.e., the empirical performance of various neural bandit algorithms. Extensive experiments are done, but the empirical results are not well-analyzed and summarized, which makes the empirical results very hard to digest. As this is a pure empirical paper, this aspect discounts the paper's impact a lot.

-------------- After Rebuttal -------------
It seems my concerns are not addressed, so i will keep my score.

---

> ### Author Response · Authors · 2022-11-18
> **Response to Reviewer qgBu**
>
> We sincerely thank the reviewer for the comments and suggestions.
>
> We will keep improving work and try to include more scenarios into our experiments to provide a comprehensive analysis of the neural bandit algorithms. In this work, we mainly focus on experiments and indeed missed necessary analysis of the results.  In order to provide a thorough analysis, we will focus on summarizing the difference among different datasets and try to quantify which algorithm performs better in what circumstances. We agree that including the recommender system scenario is necessary and we will give it a try in the future.

---

### Decision · Program_Chairs · 2023-01-20

**Decision:**

Reject

**Justification For Why Not Higher Score:**

See the 'weaknesses' mentioned above. All four reviewers rated this paper a reject, with no one considering it to be borderline. Most reviewers concerns would require substantial additional work to address.

**Justification For Why Not Lower Score:**

N/A

**Metareview: Summary, Strengths And Weaknesses:**

Popular bandit algorithms like UCB & Thompson sampling require careful quantification of uncertainty. With linear models, generalized linear models, kernel methods etc, uncertainty quantification is relatively well understood. But when neural networks are used, uncertainty quantification is much more subtle.

This paper focuses on approaches that approximate uncertainty by pretending the neural network is a linear model, either using the last layer while treating parameters of hidden layers as fixed, or using a first-order Taylor expansion (in NTK style).  Several algorithms of this type are tested empirically across several datasets. They are compared to eachother and to a linear UCB algorithm that assumes rewards are linear in whatever feature representation is used by default in the data.

*Strengths*: Careful, reproducible, empirical studies can be of great benefit to the community. Doing this well is very difficult and is not sufficiently incentivized. I WANT to accept the paper insofar as doing so would encourage more papers with similar goals.

*Weaknesses*:
1)  It is often unclear what we are meant to learn from empirical comparisons. One neural bandit algorithm outperforms the others on one dataset but, but they swap orders on another dataset. How does this depend on the properties of the dataset? Or even on the choice of tuning parameter?  Reviewer qgBu stresses this point.

2) A very limited range of methods are compared. Algorithms that fit neural networks and then perform uncertainty quantification by linearizing are compared to algorithms that perform linear regression with whatever covariates are in the dataset. [Without any feature engineering]. This area of study has some big questions around when linearizing is adequate and whether this approach provides benefits over separating representation learning from bandit style exploration. The current comparisons cannot offer insight into these issues. Reviewer Zjj4 comments on this issue. Reviewer 22TM emphasizes the lack of alternative benchmarks.

3) Limited range of datasets. Reviewer sUDB highlights that the paper 'A contextual bandit bakeoff' compared algorithms across over 500 datasets. Asa a result, the paper could identify the types of environments in which certain algorithms struggle. I worry that we set too high a bar for papers that conduct empirical comparisons of existing methods. However it is notable that the 'bakeoff' does a much better job at testing a diversity of approaches, comparing across many datasets, offering insights, and facilitation easy replication. The current submission  falls too far short.